# Differential coexistence of multiple genotypes of *Ophiocordyceps sinensis* in the stromata, ascocarps and ascospores of natural *Cordyceps sinensis*

**Yu-Ling Li**[1☯]**, Xiu-Zhang Li**[1☯]**, Yi-Sang Yao**[2☯]**, Zi-Mei Wu**[3]**, Ling Gao**[2]**, Ning-Zhi Tan**[2]**, Zhou-Qing Luo**[3¤]**, Wei-Dong Xie**[2]**, Jian-Yong Wu**[4,5]**, Jia-Shi Zhu**[1,2]***

**1** State Key Laboratory of Plateau Ecology and Agriculture, Qinghai Academy of Animal and Veterinary Sciences, Qinghai University, Xining, Qinghai, China, **2** Shenzhen Key Laboratory of Health Science and Technology, Institute of Biopharmaceutical and Health Engineering, Shenzhen International Graduate School, Tsinghua University, Shenzhen, China, **3** School of Life Sciences, Tsinghua University, Beijing, China, **4** State Key Laboratory of Chinese Medicine and Molecular Pharmacology, Shenzhen, Guangdong, China, **5** Department of Applied Biology and Chemistry Technology, The Hong Kong Polytechnic University, Hung Hom, Kowloon, Hong Kong

☯ These authors contributed equally to this work.
¤ Current address: State Key Laboratory of Cellular Stress Biology, School of Life Sciences, Faculty of Medicine and Life Sciences, Xiamen University, Xiamen, China
* zhujosh@163.com

**Data Availability Statement:** All relevant data are within the manuscript and its Supporting information files.

## Abstract

### Objective

To examine the differential occurrence of *Ophiocordyceps sinensis* genotypes in the stroma, stromal fertile portion (SFP) densely covered with numerous ascocarps, and ascospores of natural *Cordyceps sinensis*.

### Methods

Immature and mature *C. sinensis* specimens were harvested. Mature *C. sinensis* specimens were continuously cultivated in our laboratory (altitude 2,200 m). The SFPs (with ascocarps) and ascospores of *C. sinensis* were collected for microscopic and molecular analyses using species-/genotype-specific primers. Sequences of mutant genotypes of *O. sinensis* were aligned with that of Genotype #1 *Hirsutella sinensis* and compared phylogenetically using a Bayesian majority-rule method.

### Results

Fully and semiejected ascospores were collected from the same specimens. The semiejected ascospores tightly adhered to the surface of the asci as observed by the naked eye and under optical and confocal microscopies. The multicellular heterokaryotic ascospores showed uneven staining of nuclei. The immature and mature stromata, SFPs (with ascocarps) and ascospores were found to differentially contain several GC- and AT-biased genotypes of *O. sinensis*, *Samsoniella hepiali*, and an AB067719-type fungus. The genotypes

**Funding:** This research was supported by a grant from the Science and Technology Department of Qinghai Province, China, grant number 2021-SF-A4 "Study on key technologies of conservation of natural resource and industrial upgrading of Cordyceps sinensis", the major science and technology projects in Qinghai Province.

**Competing interests:** The authors have declared that no competing interests exist.

within AT-biased Cluster-A in the Bayesian tree occurred in all compartments of *C. sinensis*, but those within AT-biased Cluster-B were present in immature and mature stromata and SPFs but absent in the ascospores. Genotype #13 of *O. sinensis* was present in semi-ejected ascospores and Genotype #14 in fully ejected ascospores. GC-biased Genotypes #13–14 featured large DNA segment substitutions and genetic material recombination between the genomes of the parental fungi (*H. sinensis* and the AB067719-type fungus). These ascosporic offspring genotypes combined with varying abundances of *S. hepiali* in the 2 types of ascospores participated in the control of the development, maturation and ejection of the ascospores.

## Conclusion

Multiple genotypes of *O. sinensis* coexist differentially in the stromata, SFPs and 2 types of *C. sinensis* ascospores, along with *S. hepiali* and the AB067719-type fungus. The fungal components in different combinations and their dynamic alterations in the compartments of *C. sinensis* during maturation play symbiotic roles in the lifecycle of natural *C. sinensis*.

## Introduction

Natural *Cordyceps sinensis* is one of the most highly valued therapeutic agents in traditional Chinese medicine (TCM) and has been used for centuries in clinics as a folk medicine for "Yin-Yang" double invigoration, health maintenance, disease amelioration, and post-disease/surgery recovery [1, 2]. Modern pharmacological studies have validated the therapeutic profile and lifespan-extending properties of natural *C. sinensis* and its mycelial fermentation products [2–6]. The Chinese Pharmacopeia defines natural *C. sinensis* as an insect-fungal complex consisting of the fruiting body of *Ophiocordyceps sinensis* and the remains of Hepialidae moth larva, *i.e.*, natural *C. sinensis* ≠ *O. sinensis* fungus [7–12]. Studies have reported that the caterpillar body of *C. sinensis* contains an intact larval intestine, an intact and thick larval body wall with numerous bristles, head tissues, and fragments of other larval tissues [8–12]. The controversy surrounding the indiscriminate use of Latin names (*Cordyceps sinensis* and *Ophiocordyceps sinensis* since 2007) for the natural insect-fungal complex and multiple anamorphic-teleomorphic fungi has been addressed both in Chinese and English [7–12]. In this paper, we temporarily refer to the fungus/fungi as *O. sinensis* and continue the customary use of the name *C. sinensis* for the wild or cultivated insect-fungal complex, although this practice will likely be replaced by the discriminative use of exclusive Latin names in the future.

Natural *C. sinensis* grows only in alpine areas at altitudes above 3,000–3,500 m on the Qinghai-Tibetan Plateau and has a complex lifecycle [2, 12–14]. Its maturation stages, which have been used as a market standard for grading the quality of natural *C. sinensis*, greatly impact its mycobiota profile, metagenomic polymorphism, metatranscriptomic and proteomic expression, chemical constituent fingerprint, competitive proliferation of *Hirsutella sinensis*-like fungi, and therapeutic efficacy and potency as a natural therapeutic agent [7–10, 12, 14–28]. Mycologists have identified 22 species from 13 fungal genera in this insect-fungal complex [29], and culture-independent molecular methodologies have identified >90 fungal species spanning more than 37 genera and 12 genotypes of *O. sinensis* and demonstrated the predominance of different fungi and metagenomic fungal diversity in the stroma and caterpillar body of natural and cultivated *C. sinensis* [9, 15, 16, 18, 24, 27–40].

Wei et al. [41] hypothesized that *H. sinensis* is the sole anamorph of *O. sinensis*. This hypothesis was primarily based on 3 lines of evidence: (1) frequent isolation and mycological identification according to sporulation, conidial morphology and growth characteristics [29]; (2) the microcycle conidiation of ascospores [42–44]; and (3) systematic molecular analyses *via* internal transcribed spacer (ITS) sequencing and random molecular marker polymorphism assays [25, 31–35, 41, 45, 46]. Wei et al. [47] reported an industrial artificial cultivation project and demonstrated a mismatch between the inoculants of 3 GC-biased Genotype #1 *H. sinensis* strains and a sole teleomorphic fungus (AT-biased Genotype #4 of *O. sinensis*) in the fruiting body of cultivated *C. sinensis*. The sequences of the AT-biased genotypes of *O. sinensis* are absent in the genome of GC-biased Genotype #1 *H. sinensis* and belong to independent *O. sinensis* fungi [7–12, 24, 48, 49–54, 63, 64]. Thus, according to the fourth criterion of Koch's postulates, the species contradiction between the anamorphic inoculants and the teleomorph in cultivated products disproves the sole anamorph hypothesis for *H. sinensis* that was proposed by Wei et al. [41] of the same research group 10 years ago. The teleomorphs of *O. sinensis* found in natural *C. sinensis* specimens collected from geographically distant areas and cultivated *C. sinensis* reportedly belong to Genotypes #1, #2, #3, #4, #5 or #7 [27–28, 30–35, 41, 45, 46, 55–58], whereas those found in ascospores of *C. sinensis* reportedly belong to both GC-biased Genotype #1 and AT-biased Genotype #5 of *O. sinensis* fungi [59]. In addition to the report of species contradiction between the anamorphic inoculants (GC-biased Genotype #1) and teleomorphs (AT-biased Genotype #4) in cultivated *C. sinensis*, Wei et al. [47] identified teleomorphic Genotype #1 of *O. sinensis* in natural *C. sinensis*. Because the majority of the fungal species present in the natural world are most likely nonculturable [60–62], culture-independent molecular approaches have been widely used, resulting in the identification of 12 genotypes of *O. sinensis* and many other fungi from natural and cultivated *C. sinensis* in previous studies [9, 16–18, 24, 27, 28, 30–35, 40, 45, 47, 54–59, 63–66].

In this study, natural *C. sinensis* specimens were collected from the Hualong and Yushu areas of Qinghai Province and cultivated continuously in our laboratory in Xining city (altitude of 2,200 m). The histology of the stromal fertile portion (SFP) that was densely covered with ascocarps and ascospores of *C. sinensis* was examined under optical, confocal and fluorescence microscopes. Multiple genotypes of *O. sinensis* and other *C. sinensis*-associated fungi were profiled in the immature and mature stromata, fertile portion of the mature stroma and 2 types of ascospores of natural *C. sinensis* using species- and genotype-specific PCR primers, amplicon sequencing and cloning-based sequencing approaches.

## Materials and methods

### Reagents

Laboratory common reagents such as ethanol, sucrose, paraformaldehyde, hematoxylin, eosin, agarose and electrophoresis reagents, etc. were purchased from Beijing Bioland Technology Company. Mercuric chloride (0.1%) for surface sterilization of freshly collected *C. sinensis* specimens was a gift from the Institute of Microbiology, Chinese Academy of Sciences. *Eco*RI endonuclease is a product of New England BioLabs, United States. The Universal DNA Purification kit was a product of TIANGEN BIOTECH Company, China. The DNeasy Plant Mini Kit was a product of Qiagen Company, Germany. The Gel Extraction Kit was a product of Omega Bio-Tek, United States. The *Taq* PCR reagent kit and Vector NTI Advance 9 software were purchased from Invitrogen, United States. Calmodulin was a product of Abicom (Shanghai, China).

## Collection of *C. sinensis* specimens and ascospores

Immature *C. sinensis* specimens were purchased from local markets from the Hualong (located at 36˚13'N, 102˚19'E) and Yushu areas (located at 33˚01'N, 96˚48'E) of Qinghai Province of China (3800–4600 m' altitude) in mid-May and characterized by a plump caterpillar body and very short stroma (1.0–2.0 cm) [19, 27]. Mature *C. sinensis* specimens were collected in mid-June and characterized by a plump caterpillar body and long stroma (>5.0 cm) and by the formation of an expanded fertile portion close to the stromal tip, which was densely covered with ascocarps (Fig 1). Governmental permission was not required for *C. sinensis* purchases in local markets, and the collections of *C. sinensis* specimens from sales by local farmers fall under the governmental regulations for traditional Chinese herbal products.

The specimens were washed thoroughly on site in running water with gentle brushing, soaked in 0.1% mercuric chloride for 10 min for surface sterilization and washed 3 times with sterile water. The thoroughly cleaned specimens were immediately frozen in liquid nitrogen on site and kept frozen during transportation to the laboratory and during storage prior to further processing [19, 27].

Some of the mature *C. sinensis* specimens were harvested along with the outer mycelial cortices and soil surrounding the caterpillar body and replanted in paper cups in soil obtained from *C. sinensis* production areas (Fig 1A) and were cultivated in our laboratory (altitude 2,200 m) in Xining City, Qinghai Province of China [67, 68]. Because of the phototropism of natural *C. sinensis*, we kept the windows fully open, allowing sufficient sunshine and a natural plateau breeze blowing over the cultivated specimens in the paper cups. The room temperature was maintained naturally, fluctuating with the lowest temperature at 18–19˚C during the night and the highest temperature at 22–23˚C in the early afternoon. The humidity of our laboratory was maintained by spraying of water using an atomizer twice a day in the morning and evening.

Fully ejected ascospores of *C. sinensis* were collected using double layers of autoclaved weighing paper (Fig 1B). During massive ascospore ejection, numerous ascospores adhered to the outer surface of asci, as shown in Fig 1C after removing the upper layer of autoclaved weighing papers for collection of the fully ejected ascospores, and failed to be brushed away using an autoclaved brush; hence, these ascospores were instead gently scratched off using a disinfected inoculation shovel or ring and referred to as semiejected ascospores.

The 2 types of ascospores were cleaned by 2 washes with 10% and 20% sucrose solutions and 10-min centrifugation at 1,000 rpm (desktop centrifuge, Eppendorf, Germany); the supernatant was discarded after each centrifugation. The pellets (ascospores) were subsequently washed with 50% sucrose solution and centrifuged for 30 min, and the ascospores that floated on the liquid were collected [67]. The fully and semiejected ascospores were stored in a -80˚C freezer prior to further processing.

## Histological examination of the SFP, ascocarps and ascospores of *C. sinensis*

The fully ejected ascospores of *C. sinensis* were diluted in normal saline, placed on a glass slide, and air-dried for histological examination under an optical microscope (Model BX51, OLYMPUS, Japan) without staining. The ascospores were fixed on a glass slide with 4% paraformaldehyde for 1 h, incubated in 0.01% calmodulin for 5 min for the visualization of septa, washed 3 times with PBS, and observed under a fluorescence microscope with UV epi-illumination (Model XZ51, OLYMPUS, Japan) [68].

The mature *C. sinensis* stromata collected during the massive ejection of ascospores were immersed in 10% formalin for fixation and subjected to dehydration in 50%, 70% and 95%

(A)

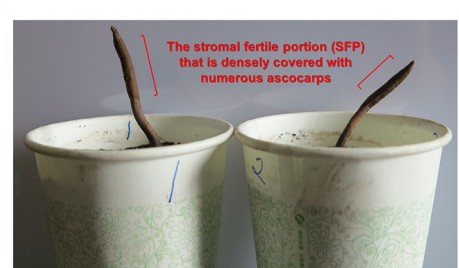

(B)

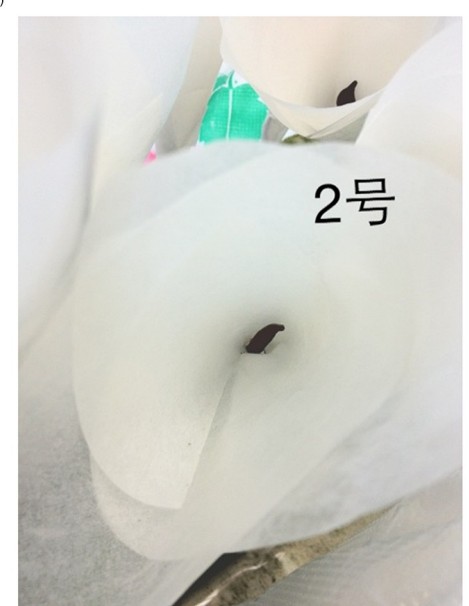

(C)

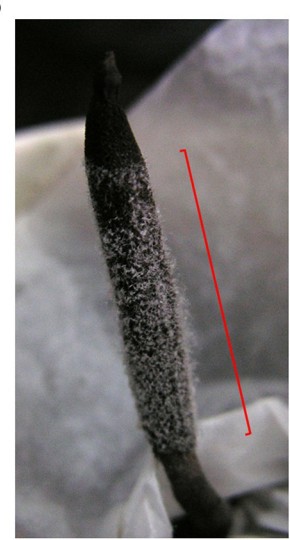

**Fig 1. Cultivation of mature *C. sinensis* specimens in paper cups and collection of ascospores.** Mature *C. sinensis* specimens were cultivated in our Xining laboratory (altitude of 2,200 m) (Fig 1A). The fully ejected ascospores were collected using the double layers of autoclaved weighing papers (Fig 1B). Numerous semiejected ascospores adhere to the outer surface of an ascus (Fig 1C, after removing the upper layer of autoclaved weighing papers for collection of the fully ejected ascospores) during the massive ejection of ascospores. The stromal fertile portion (SFP) densely covered with numerous ascocarps is labeled with "]".

ethanol for 1 h each [69]. The SFP tissues (densely covered with ascocarps) were embedded in paraffin and sliced to 5-μm thickness (Model TN7000 Microtome, Tanner Scientific, Germany). The ascus slices were stained with hematoxylin-eosin and observed under optical and confocal microscopes (Model Primo Star and Model LSM780, ZEISS, Germany).

## Extraction and preparation of genomic DNA

The frozen test samples, including the stroma, SFP (densely covered with ascocarps), and the 2 types of ascospores of *C. sinensis*, were individually ground to a powder in liquid nitrogen. The mycelia of pure fungi, including *H. sinensis* and *Samsoniella hepiali* (synonym *Paecilomyces hepiali* [70]) (gifts from Prof. Guo Y-L [26]), *Geomyces pannorum*, *Penicillium chrysogenum* and *Pseudogymnoascus roseus* (gifts from Prof. Zhang Y-J), and *Tolypocladium sinensis* (provided by Prof. Wu J-Y), were also individually ground to powder in liquid nitrogen; these fungi have been frequently detected in natural *C. sinensis*, and some of them reportedly show differential dominance in the stroma and caterpillar body of *C. sinensis* [23, 26–29, 34, 36–39, 71–73]. Genomic DNA was individually extracted from these powder samples using the DNeasy Plant Mini Kit (Qiagen) according to the manufacturer's manual [19, 26].

## Universal primers and genotype- and species-specific primers

Table 1 lists the sequences of the *IST5/ITS4* universal primers, fungal species-specific primers and the genotype-specific primers for GC- and AT-biased genotypes of *O. sinensis*. The

**Table 1. ITS5/ITS4 universal primers and genotype- and species-specific primers used for the PCR amplification and sequencing of ITS segments.**

| Primer | Direction | Primer sequence |
|---|---|---|
| **Universal Primers** | | |
| *ITS5* | Forward | GGAAGTAAAAGTCGTAACAAGG |
| *ITS4* | Reverse | TCCTCCGCTTATTGATATGC |
| **Genotype-specific primers designed based on AB067721 of the GC-biased Genotype #1 *H. sinensis*** | | |
| *Hsprp1* | Forward | ATTATCGAGTCACCACTCCCAAACCCCC |
| *Hsprp2* | Reverse | ATTTGCTTGCTTCTTGACTGAGAGATGCC |
| *Hsprp3* | Reverse | CGAGGTTCTCAGCGAGCTACT |
| **Genotype-specific primers designed based on AB067744 and AB067740 of the AT-biased *O. sinensis* genotypes** | | |
| *HsATp1* | Forward | AAGGTCTCCGTTAGTAAACT |
| *HsATp2* | Reverse | GGGGCTCGAGGGTTAAGATA |
| *HsATp3* | Reverse | GGGGCTTAAGGGTTAAGGTA |
| **Species-specific primers designed based on DQ189229 of *Geomyces pannorum* and AY608922 of *Pseudogymnoascus roseus*** | | |
| *Prp2* | Forward | ATTACACTTTGTTGCTTTGGCA |
| *Prp5* | Reverse | GCTGGCGAGCACACGACCGGACCT |
| **Species-specific primers designed based on DQ336710 of *Penicillium chrysogenum*** | | |
| *Pcp3* | Forward | GAGGGCCCTCTGGGTCCAACC |
| *Pcp7* | Reverse | CCCCATACGCTCGAGGACC |
| **Species-specific primers designed based on EF555097 of *Samoneilla hepiali* (≡*Paecilomyces hepiali*)** | | |
| *Php4* | Forward | GTATCTTCTGAATCCGCCGCAAGGC |
| *Php6* | Reverse | AACGTTCAGAAGTCGGGGGTTTTAC |
| **Species-specific primers designed based on DQ097715 of *Tolypocladium sinensis*** | | |
| *Tsp1* | Forward | GACCGCCCCGGCGCCCTCG |
| *Tsp3* | Reverse | TGACCGTCTCCGCGCT |
| **Primers used for PCR2.1 vector clone sequencing** | | |
| *M13F* | Forward | TGTAAAACGACGGCGT |
| *M13R* | Reverse | CAGGAAACAGCTATCC |

positions of the genotype-specific primers are shown in Fig 2. All primers were synthesized by Invitrogen Beijing Lab. or Beijing Bomaide Technology Co.

Among multiple pairs of species-specific primers that were designed for *G. pannorum*, *H. sinensis*, *P. chrysogenum*, *P. roseus*, *S. hepiali* and *T. sinensis*, the following primer pairs were selected according to their specificity and amplification efficiency as examined through PCR amplification and sequencing using the genomic DNA templates isolated from the fungal mycelia (listed in Table 1): *Prp2/Prp5* for *G. pannorum* and *P. roseus*, *Pcp3/Pcp7* for *P. chrysogenum*, *Php4/Php6* for *S. hepiali*, and *Tsp1/Tsp3* for *T. sinensis*. The ITS sequences of *P. roseus* (AY608922) and *G. pannorum* (JF320819 and DQ189229) are 98–99% homologous and were amplified using the same pair of primers, *Prp2/Prp5* (Table 1).

## PCR protocol for the amplification of ITS segments

The genomic DNA templates and aforementioned universal, genotype- and species-specific primers were used in PCR assays using a PCR instrument (Bio-Rad, United States) and *Taq* PCR reagent kit (Invitrogen) to amplify the ITS1-5.8S-ITS2 segments using the following touch-down protocol: (1) 95˚C for 5 min; (2) 36 cycles of 95˚C for 30 sec, annealing temperature for 30 sec (the annealing temperature was initially set to 70˚C and decreased stepwise by 0.3˚C in each cycle), and 72˚C for 1 min; (3) 72˚C for 10 min and final incubation at 4˚C [26, 27]. The PCR amplicons were examined by agarose gel electrophoresis and sequencing.

## Amplicon sequencing, cloning-based sequencing and sequence analysis

Each of the targeted PCR amplicons obtained from the aforementioned genomic DNA templates was recovered from agarose gels using a Gel Extraction Kit (Omega Bio-Tek) and purified using a Universal DNA Purification kit (TIANGEN) according to the manufacturer's manuals [26, 27]. The purified amplicons were subjected to *Eco*RI endonuclease (BioLabs) digestion, which specifically digested the GC-biased genotypes of *O. sinensis* but not the AT-biased genotypes (*cf*. the underlined "GAATTC" site in green shown in Fig 2) and analyzed by agarose gel electrophoresis. The purified amplicons were sequenced either directly or after cloning by Invitrogen Beijing Lab.

For cloning, the amplicon was inserted into a PCR2.1 vector (SHQIMBIO, Shanghai, China), which was then transfected into DH5α cells, and the cells were coated on agar containing 100 μg/mL ampicillin or kanamycin in Petri dishes. The Petri dishes were cultured at 37˚C, allowing the growth of cells; 30 white colonies were selected per dish, transferred to liquid culture medium and grown at 37˚C in a shaking incubator. The expanded clones were sequenced using the M13F/M13R primers (*cf*. Table 1) (Invitrogen or Beijing Bomaide Technology Co.). The sequences were analyzed using Vector NTI Advance 9 software (Invitrogen) [26, 27].

## Phylogenetic analysis of ITS sequences

All 17 genotypes of *O. sinensis* available in the GenBank database and obtained from this study were analyzed by Nanjing Genepioneer Biotechnologies Co. to reveal their phylogenetic relationships. A Bayesian majority-rule consensus tree was inferred using MrBayes v3.2.7a software (the Markov chain Monte Carlo [MCMC] algorithm; http://nbisweden.github.io/MrBayes/) with a sampling frequency of $10^3$ iterations after discarding the first 25% of samples from a total of $1.1 \times 10^8$ iterations [74].

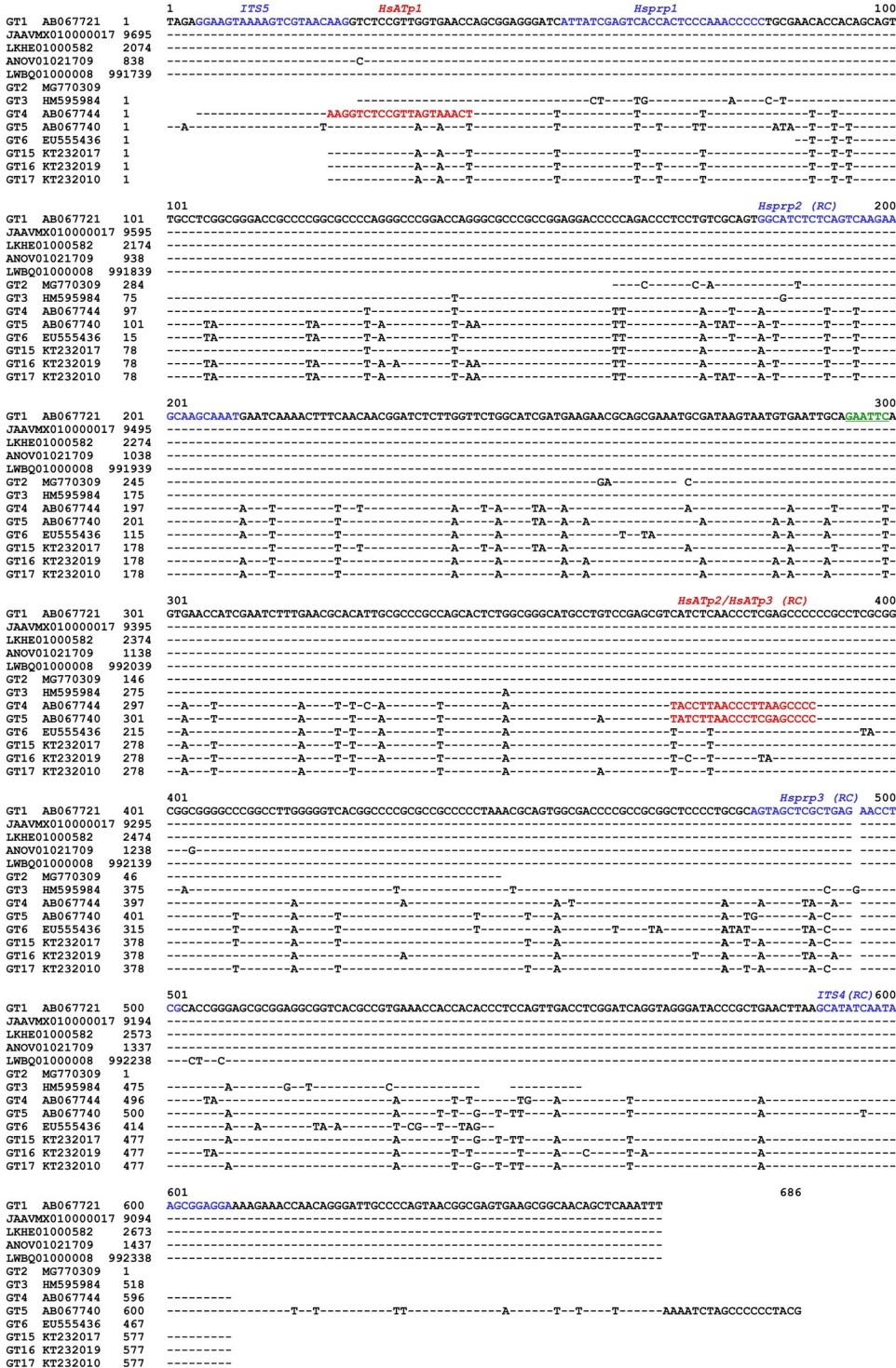

**Fig 2. Alignment of the ITS sequences of GC- and AT-biased genotypes of *O. sinensis* with multiple transition point mutations.** GT represents genotype. Genotypes #1–3 are GC-biased *O. sinensis* genotypes, and Genotypes #4–6 and #15–17 are AT-biased *O. sinensis* genotypes. The sequence segments shown in blue correspond to the primers designed based on the sequences of GC-biased genotypes, and those in red correspond to primers designed based on the sequences of AT-biased genotypes. The underlined "GAATTC" site shown in green is the enzymatic site of the *Eco*RI endonuclease, and it is present in the GC-biased sequences at nucleotides 294–299 in Genotype #1 but absent in the AT-biased sequences due to a single-base mutation (GAATTT). "(RC)" denotes the reverse complement sequence of the primers; "-" represents identical bases; and spaces indicate unmatched sequence gaps.

## Results

### Two types of *C. sinensis* ascospores

To mimic the wild environment of the Qinghai-Tibetan Plateau, mature *C. sinensis* specimens were cultivated in paper cups in our Xining laboratory (altitude 2,200 m) (Fig 1A). White ascospores started to become visible on the surface of asci after approximately one week of cultivation. The fully ejected ascospores were collected on the double layers of autoclaved weighing paper (Fig 1B), and the numerous semiejected ascospores that tightly adhered to the outer surface of asci (Fig 1C after removing the upper layer of autoclaved weighing papers for collection of the fully ejected ascospores) were collected by gentle scraping using a disinfected inoculation shovel or ring [67].

### Microscopy observations of the SFPs, ascocarps and ascospores of *C. sinensis*

Fig 3 shows the histology of the fully ejected *C. sinensis* ascospores without staining (upper panel; 40x) and after staining with calmodulin to visualize the septa of the multicellular structure of the ascospores (lower panel; 400x).

Fig 4A shows a confocal image (bar, 500 μm) of a transverse section of the fertile portion of the *C. sinensis* stroma, which is densely covered with multiple ascocarps. Fig 4B shows an optical microscopy image (10x) of several ascocarps, and Fig 4C shows a close-up optical microscopy image (40x) of an ascocarp. Ascocarps were stained with hematoxylin-eosin containing multiple ascospores, which revealed uneven staining of nuclei (dark blue–purple), consistent with the multicellular heterokaryosis of the *C. sinensis* ascospores reported by Bushley et al.

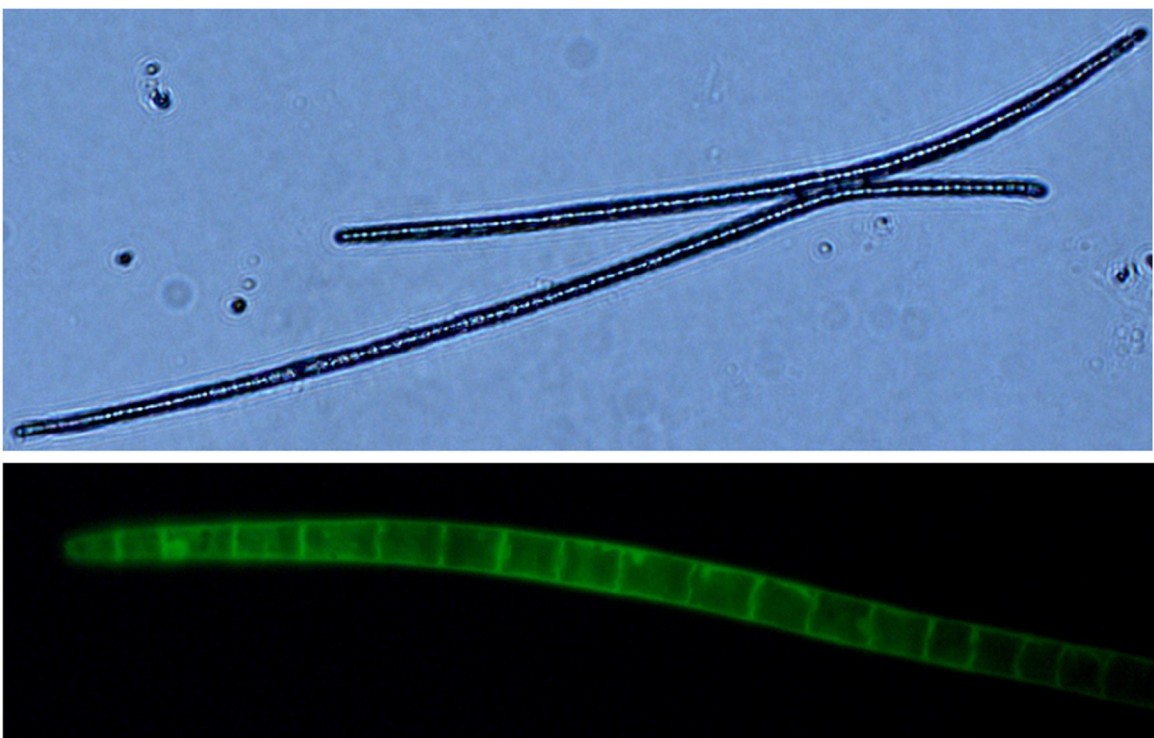

**Fig 3. Microscopy images of fully ejected ascospores of *C. sinensis* without staining (upper panel; 40x) or after staining with 0.01% calmodulin for visualization of the septa of multicellular ascospores (lower panel; 400x).**

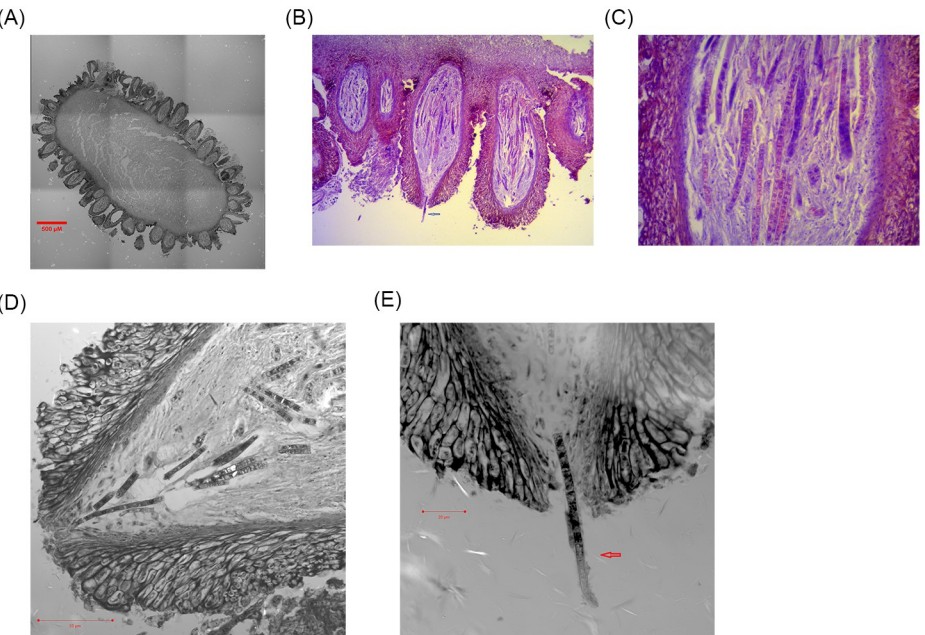

**Fig 4. Microscopy images of the SFP, ascocarps and ascospores of *C. sinensis*.** Fig 4A is a confocal image of a transverse section of the SFP (bar, 500 μm). Fig 4B is an optical microscopic image (10x) of several *C. sinensis* ascocarps stained with hematoxylin-eosin. Fig 4C is a close-up optical image (40x) of an ascocarp stained with hematoxylin-eosin. Fig 4D and 4E are close-up confocal images showing ascospores gathering toward the opening of the perithecium (Fig 4D; bar, 50 μm) and a semiejected ascospore hanging out of the opening of the perithecium (Fig 4E; bar, 20 μm). Arrows in Fig 4B and 4E indicate semiejected ascospores.

[75]. Fig 4D and 4E provide confocal images close to the opening of the perithecia, showing ascospores gathering toward the opening of the perithecium (Fig 4D; bar, 50 μm) and a semie-jected ascospore hanging out and adhered to the opening of the perithecium (Fig 4E; bar, 20 μm). Arrows in Fig 4B and 4E indicate semiejected ascospores.

## ITS sequences amplified using universal primers

ITS sequences were amplified from the immature and mature stroma, SFP (with ascocarps), and fully and semiejected ascospores of *C. sinensis* using the universal primers *ITS5/ITS4*. The targeted amplicons of 630$^+$ bp are 99% homologous to sequence AB067721 of *H. sinensis* (Genotype #1 of *O. sinensis*) [18, 23–24, 26–27, 31, 33, 35, 41, 45].

## ITS sequences amplified using species-specific primers

ITS sequences were amplified from the genomic DNA templates obtained from immature and mature stroma, SFP (with ascocarps), and fully and semiejected ascospores of *C. sinensis* using the *S. hepiali*-specific primers *Php4/Php6*. The targeted amplicons of 460$^+$ bp in the sample lanes shown in Fig 5 were recovered and sequenced, and the sequences were 100% homolo-gous to the EF555097 sequence of *S. hepiali* [26]. As determined through PCR amplification of the same quantity of genomic DNA, significantly lower abundance of the moieties of the ITS amplicons was obtained after amplification from genomic DNA of the fully ejected ascospores (lanes 1 and 3 of Fig 5) than after amplification of genomic DNA of the semiejected ascospores (lanes 2 and 4).

In other PCR experiments, the ITS sequences of various fungi were amplified from the SFP (with ascocarps) and fully and semiejected ascospores of *C. sinensis* using the following

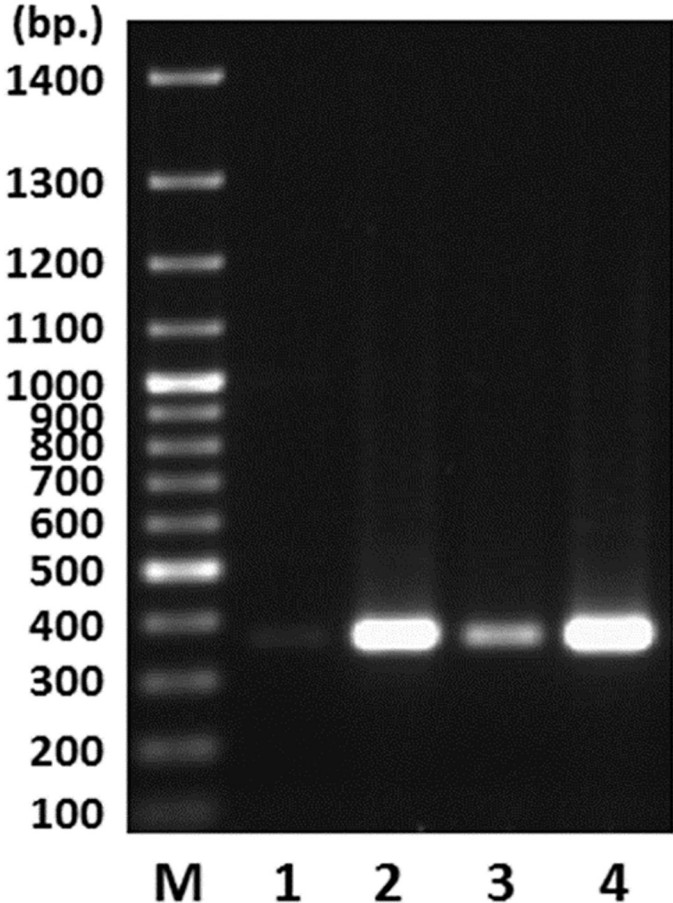

**Fig 5. Agarose gel electrophoresis of the PCR amplicons obtained from genomic DNA of the fully and semiejected ascospores of *C. sinensis* using the *Samsoniella hepiali*-specific Php4/Php6 primers.** Lane M shows the molecular weight standard. Lanes 1 and 3 display the amplicon moieties amplified from the genomic DNA of the fully ejected ascospores. Lanes 2 and 4 show the amplicons amplified from the genomic DNA of the semiejected ascospores.

species-specific primer pairs (*cf*. Table 1): *Prp2/Prp5* for *G. pannorum* (JF320819) and *P. roseus* (AY608922), *Pcp3/Pcp7* for *P. chrysogenum* (DQ336710), and *Tsp1/Tsp3* for *T. sinensis* (DQ097715) (*cf*. Table 1). Amplicon sequencing did not detect the sequences of *G. pannorum*, *P. roseus*, or *T. sinensis*. Agarose gel electrophoresis showed that the amplicons obtained using the primers *Pcp3/Pcp7* migrated at a speed similar to the positive control moiety of *P. chrysogenum*, but sequencing analysis did not reveal the ITS sequence of *P. chrysogenum*. Thus, species-specific PCR primers failed to detect the ITS sequences of *G. pannorum*, *P. chrysogenum*, *P. roseus*, or *T. sinensis* in the SFP (with ascocarps) and fully and semiejected ascospores of *C. sinensis*.

## Profiling of multiple *O. sinensis* genotypes in the compartments of *C. sinensis* using a cloning-based sequencing approach

Multiple genotypes of *O. sinensis* were further examined using genomic DNA obtained from the immature and mature stromata, SFP (with ascocarps), and 2 types of ascospores of *C. sinensis*. Multiple genotype-specific primers were designed and tested for specificity and amplification efficiency, and 5 pairs were selected for amplification of the ITS1-5.8S-ITS2 sequences of the GC- and AT-biased *O. sinensis* genotypes (*cf*. Table 1). The universal primers

**Table 2. Differential occurrence of multiple genotypes of *O. sinensis*, *S. hepiali* and AB067719-type fungus in the compartments of natural *C. sinensis*.**

| Genotype | Representative sequence | Stroma | | SFP (with ascocarps) | Ascospores | |
| --- | --- | --- | --- | --- | --- | --- |
| | | Immature | Mature | | Fully ejected | Semiejected |
| #1 | AB067721 | √ | √ | √ | √ | √ |
| #2 | MG770309 | √ | √ | | | |
| #4 | AB067744 | √ | √ | √ | | |
| #5 | AB067740 | √ | √ | √ | √ | √ |
| #6 | EU555436 | √ | √ | √ | √ | |
| #13 | KT339190 | | | | | √ |
| #14 | KT339178 | | | | √ | |
| #15 | KT232017 | √ | √ | √ | | |
| #16 | KT232019 | √ | | | √ | |
| #17 | KT232010 | √ | | | | |
| *S. hepiali* | EF555097 | √ | √ | √ | √ | √ |
| AB067719-type fungus | AB067719 | √ | √ | √ | √ | √ |

*ITS5/ITS4* and the genotype-specific primer pairs *Hsprp1/Hsprp2*, *Hsprp1/Hsprp3*, or *Hsprp2/Hsprp3* were highly homologous to GC-biased Genotypes #1–2 and/or #7. The primer design did not consider GC-biased Genotypes #3 and #8–12 because the sequences were obtained from *C. sinensis* specimens collected from Nyingchi of Tibet and the Indian-Nepal side of the *C. sinensis* production areas, which are geographically distant from the sample collection locations in Qinghai Province used for our study. The genotype-specific primer pairs *HsATp1/ITS4*, *HsATp1/HsATp2*, and *HsATp1/HsATp3* were selected to amplify the AT-biased sequences of *O. sinensis* (*cf*. Table 1 and Fig 2). The amplification specificity and efficiency of the primer pairings have been previously tested repeatedly for the genotypes of *O. sinensis*. The PCR amplicons were subjected to cloning-based sequencing to explore the multiple mutant genotypes.

The primer pair *HsATp1/ITS4* is highly homologous to almost all AT-biased genotype sequences (100%/100%) but slightly less homologous to Genotype #5 (100%/95%) and AB067719-type sequences (95%/100%). The primer pair *HsATp1/HsATp2* is highly homologous to sequences of Genotypes #5, #15 and #17 (100%/100%) but less specific for Genotypes #4 and #16 (100%/80%). The primer pair *HsATp1/HsATp3* is highly homologous to Genotype #4 and #16 sequences (100%/100%) but less specific for Genotypes #5, #15 and #17 sequences (100%/85%). The homology of the primer *HsATp1* to the Genotype #6 sequence is unknown because all Genotype #6 sequences available in GenBank are short, probably due to the secondary structure/conformation within the ITS1 sequence close to its 5' end and the ITS2 sequence close to its 3' end. The primer *HsATp2* is 100% homologous to sequence EU555436 of Genotype #6, but *HsATp3* is less specific (85%) for EU555436. Thus, based on pretest and formal test results of the primer pairs in terms of the amplification specificity and efficiency, using all of these primer pairs in multiple PCR runs is essential for ensuring the successful amplification of all known genotypes of *O. sinensis* existing in the stroma, SFP (with ascocarps), and ascospores of *C. sinensis*.

Table 2 lists the differential occurrence of the *O. sinensis* genotypes with numerous transition point mutations detected in the immature and mature stromata, SFP (with ascocarps), and the 2 types of ascospores of *C. sinensis*. GC-biased Genotype #1 *H. sinensis* was identified in all compartments of *C. sinensis*. GC-biased Genotype #2 of *O. sinensis* was detected in both immature and mature stromata [27, 28].

**Table 3. Sequence similarities of the ITS1, 5.8S and ITS2 segments of Genotype #1 and AB067719-type sequences compared with the multiple genotypes of *O. sinensis*.**

| Genotype and representative sequence | ITS1 | 5.8S gene | ITS2 | ITS1-5.8S-ITS2 (excluding the 18S and 28S segments) |
|---|---|---|---|---|
| | Genotype #1 (AB067721) *vs.* sequences of other genotypes | | | |
| #2 MG770309 | 83.6% | 97.4% | 100% | 94.7% |
| #3 HM595984 | 94.3% | 99.4% | 93.0% | 95.5% |
| #4 AB067744 | 90.6% | 85.3% | 89.2% | 88.4% |
| #5 AB067740 | 80.5% | 86.5% | 89.2% | 85.5% |
| #6 EU555436 | 84.7% | 87.8% | 85.8% | 86.0% |
| #7 AJ488254 | 93.2% | 98.7% | 89.4% | 93.9% |
| #8 GU246286 | 86.2% | 94.8% | 87.9% | 89.6% |
| #9 GU246288 | 96.3% | 98.7% | 91.5% | 95.3% |
| #10 GU246287 | 86.2% | 92.9% | 72.4% | 83.2% |
| #11 JQ695935 | 94.3% | 100% | 55.1% | 81.6% |
| #12 GU246296 | 99.4% | 99.4% | 87.0% | 94.9% |
| #13 KT339190 | **100%** | 94.8% | 64.2% | 86.3% |
| #14 KT339178 | 67.9% | 94.9% | **100%** | 87.7% |
| #15 KT232017 | 91.8% | 86.5% | 91.7% | 89.9% |
| #16 KT232019 | 84.3% | 88.2% | 90.7% | 87.3% |
| #17 KT232010 | 83.0% | 88.5% | 91.7% | 87.7% |
| | AB067719 (Group E) *vs.* sequences of Genotypes #13–14 | | | |
| #13 KT339190 | 71.5% | **100%** | **99%** | 88.2% |
| #14 KT339178 | **100%** | **100%** | 71.5% | 89.2% |

Zhang et al. [55] and Cheng et al. [56] studied *C. sinensis* specimens collected from the Nyingchi area in Tibet and detected variable *H. sinensis* sequences corresponding to Genotype #3 of *O. sinensis* [7–12]. Chen et al. [58] detected a variable *H. sinensis* sequence AJ488254 (Genotype #7 of *O. sinensis*) in the stroma of a *C. sinensis* specimen (#H1023) collected from Qinghai Province of China, but Genotype #1 *H. sinensis* AJ488255 in the caterpillar body of the same specimen. In the current study, however, GC-biased Genotypes #3 and #7–12 were not detected in any compartments of natural *C. sinensis* collected from the Hualong and Yushu areas of Qinghai Province.

The cloning-based ITS amplicon sequencing approach showed the differential occurrence of the AT-biased sequences of Genotypes #4–6 and #15–17 of *O. sinensis* in the immature and mature stromata, SFP (with ascocarps), and 2 types of ascospores of *C. sinensis* (*cf.* Table 2). Table 2 also shows that *S. hepiali* and the AB067719-type fungus coexisted in the stromata, SFP (with ascocarps), and ascospores of *C. sinensis*.

Cloning-based amplicon sequencing also detected Genotypes #13–14 of *O. sinensis*. The Genotype #13 KT339190 sequence was detected in semiejected ascospores, whereas the Genotype #14 KT339178 sequence was detected in fully ejected ascospores (*cf.* Table 2). Table 3 compares the ITS1, 5.8S gene, and ITS2 segment sequences of the mutant genotypes with those of Genotype #1 *H. sinensis* (Group-A [66]) and the AB067719-type Group-E fungus. ITS1 of KT339190 and ITS2 of KT339178 are 100% homologous to those of Genotype #1 *H. sinensis* (AB067721). The 5.8S-ITS2 segments of KT339190 and ITS1-5.8S segments of KT339178 are 99–100% homologous to those of the AB067719-type fungus but 64.2–94.9% similar to those of the AB067721 of Genotype #1. As shown in Table 3 and Fig 6, the *O. sinensis* offspring Genotypes #13–14 resulted from large DNA segment reciprocal substitutions and

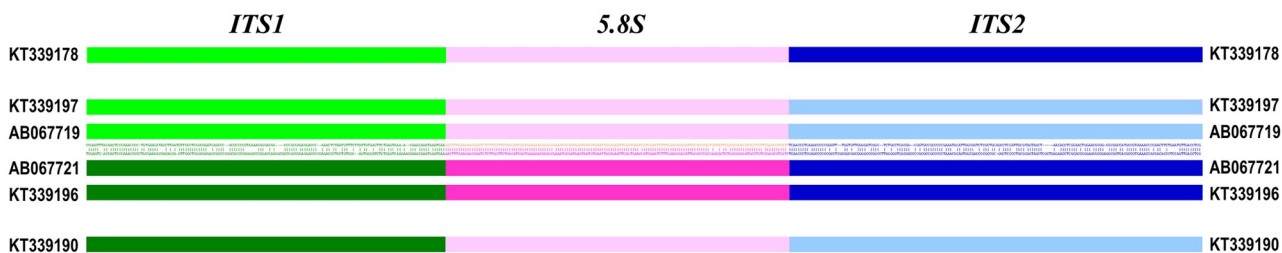

**Fig 6. Schematic representation of the ITS segment sequences of the parental fungi (*H. sinensis* and the AB067719-type fungus) and *O. sinensis* offspring Genotypes #13–14.** The green bars indicate the ITS1 segment; the pink bars refer to the 5.8S gene; the blue bars represent the ITS2 segment. AB067719 [32] and KT339197 discovered in the current study represent the AB067719-type Group-E fungus and are shown with lighter bars. AB067721 [32] and KT339196 discovered in the current study represent Genotype #1 *H. sinensis* (Group-A by [66]) and are shown with darker bars. Alignment of the AB067719 and AB067721 sequences is shown between the lighter bars for AB067719 and the darker bars for AB067721. KT339190 and KT339178 represent *O. sinensis* offspring Genotypes #13 and #14, respectively, showing large DNA segment reciprocal substitutions and genetic material recombination between the genomes of the 2 parental fungi, *H. sinensis* and the AB067719-type fungus.

genetic material recombination between the genomes of the 2 parental fungi, Group-A Genotype #1 *H. sinensis* and the AB067719-type Group-E fungus.

AB067719-type fungus in natural *C. sinensis* was grouped as a Group-E fungus by Stensrud et al. [66], unlike *O. sinensis* of Groups A–C. This fungus and 13 other sequences were annotated in the GenBank database as "*C. sinensis*" or "*O. sinensis*" under GenBank taxon 72228 for *C. sinensis* and *O. sinensis*. GenBank also collected more than 900 sequences highly homologous to AB067719, including *Alternaria* sp., *Ascomycota* sp., *Aspergillus* sp., *Avena* sp., *Berberis* sp., *Colletotrichum* sp., *Cordyceps* sp., *Cyanonectria* sp., *Dikarya* sp., *Fusarium* sp., *Gibberella* sp., *Hypocreales* sp., *Juglans* sp., *Lachnum* sp., *Nectria* sp., *Nectriaceae* sp., *Neonectria* sp., *Penicillium* sp., and many uncultured endophytic fungi. Thus, the identity of AB067719-type fungus in natural *C. sinensis* needs to be further determined through culture-dependent approaches and multigene and whole-genome sequencing of a purified AB067719-type fungus.

## Phylogenetic relationship of the genotypes of *O. sinensis*

Kinjo & Zang [32] and Stensrud et al. [66] discussed the phylogenetic relationships of Genotype #1 (Group-A) and Genotypes #4–5 (Groups B–C) of *O. sinensis*. Further phylogenetic analysis of mutant genotypes #1–12 of *O. sinensis*, which share the same *H. sinensis*-like morphological and growth characteristics, reflected the research progress that had been achieved at the time [7–12, 16–19, 25, 27, 28, 55–59]. In this study, we found additional AT-biased Genotypes #15–17 and GC-biased Genotypes #13–14 that show large DNA segment reciprocal substitutions and genetic material recombination (*cf*. Tables 2 and 3 and Fig 6). The sequences of all 17 *O. sinensis* genotypes were subjected to phylogenetic analysis using a Bayesian algorithm (Fig 7).

GC-biased genotypes of *O. sinensis* are indicated in blue alongside the tree in Fig 7, including Genotypes #1–3 and #8–14 and the ITS sequences from 5 whole-genome sequences of *H. sinensis* strains 1229, CC1406-203, Co18, IOZ07 and ZJB12195.

The AT-biased genotypes of *O. sinensis* are indicated in red alongside the tree and cluster into a clade that contains 2 branches. Cluster-A includes Genotypes #5–6 and #16–17, and Cluster-B includes Genotypes #4 and #15.

The AB067719-type Group-E fungus is side-noted in purple in Fig 7 as an outgroup control and is located outside of the blue GC-biased branch and the red AT-biased genotype clade.

Genotype #13 formed a phylogenetic leaf clustered close to the GC-biased branch. This leaf was distant from one of its parental fungi, Group-A Genotype #1 *H. sinensis*, but farther away from another parental fungus, the AB067719-type Group-E fungus.

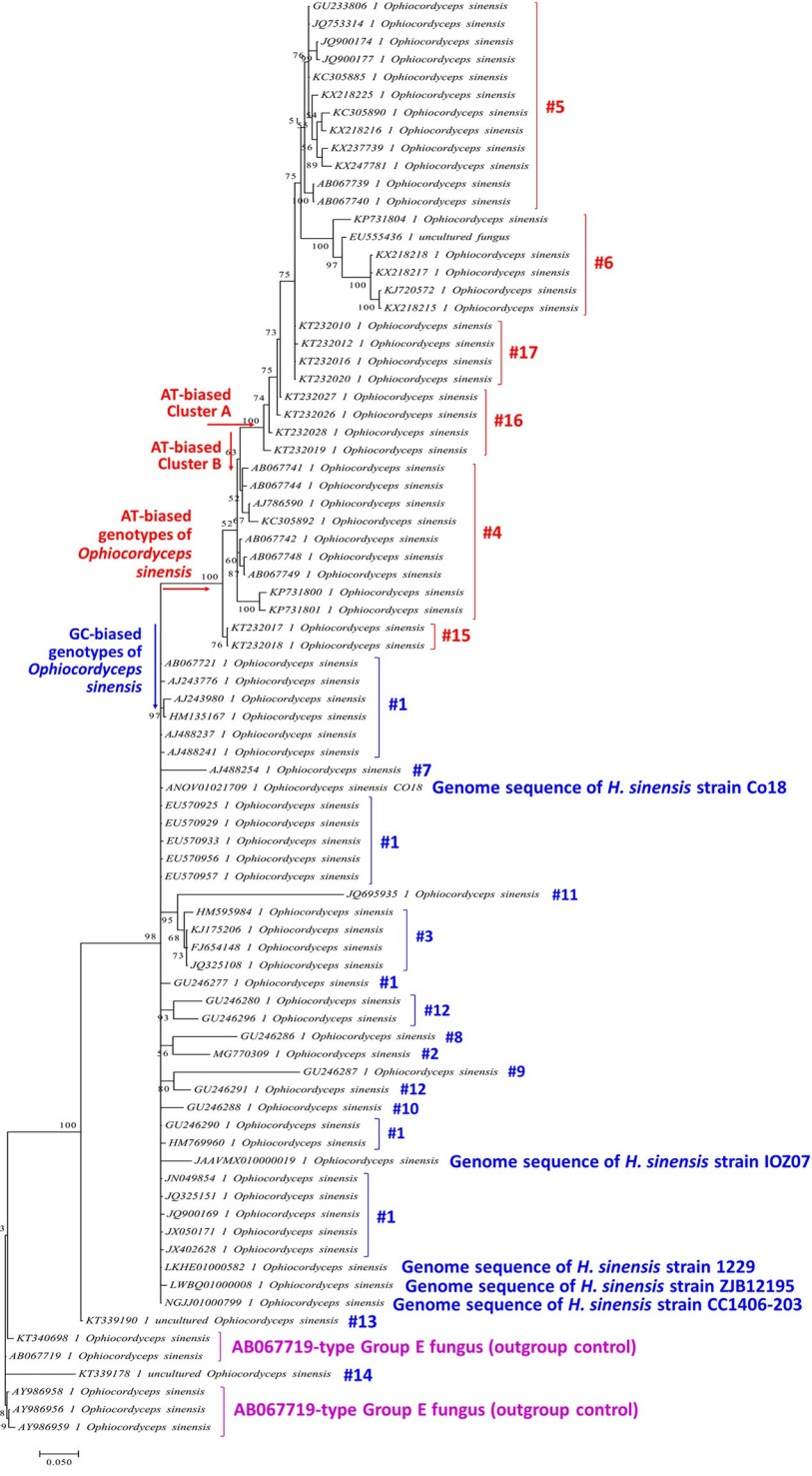

**Fig 7. Bayesian phylogenetic analysis of multiple genotypes of *O. sinensis*.** Five ITS sequences of the whole genomes (ANOV01021709, LKHE01000582, LWBQ01000008, JAAVMX010000002 and JAAVMX010000019) of *H. sinensis* strains (Co18, 1229, ZJB12195 and IOZ07) and 71 ITS sequences of 17 genotypes of *O. sinensis* were analyzed. The Bayesian majority-rule consensus tree was inferred using MrBayes v3.2.7a software (Markov chain Monte Carlo [MCMC] algorithm) [75]. GC-biased Genotypes #1–3 and #7–14 of *O. sinensis* are indicated in blue alongside the tree, and the AT-biased Genotypes #4–6 and #15–17 of *O. sinensis* are indicated in red alongside the tree. The AB067719-type Group-E sequences (5 sequences) are indicated in purple alongside the tree as an outgroup control.

Genotype #14 formed another phylogenetic leaf at a position closer to the parental AB067719-type Group-E fungus than to another parental fungus, Group-A Genotype #1 *H. sinensis*.

GC–biased Genotype #1 and AT–biased Genotypes #5–6 and #16, as well as *S. hepiali* and the AB067719-type fungus, were detected in the ascospores of *C. sinensis*. Genotypes #5–6 and #16 were located within AT-biased Cluster-A in the Bayesian tree (Fig 7). Genotypes #4 and #15 form AT-biased Cluster-B in the Bayesian tree and were detected in the stroma and SFP (with ascocarps) but not in the ascospores of *C. sinensis* (*cf*. Table 2, Fig 7).

## Discussion

### Differential occurrence of multiple genotypes of *O. sinensis* and dynamic alterations of the genotypes during *C. sinensis* maturation

In addition to the 12 *O. sinensis* genotypes previously discovered [15, 16, 18, 24, 27, 28, 30–32, 47, 55–58, 64, 66], the current study identified 5 new genotypes of *O. sinensis*: AT-biased Genotypes #15–17 with multiple transition point mutations and GC-biased Genotypes #13–14 that exhibit the characteristics of large DNA segment reciprocal substitutions and genetic material recombination in natural *C. sinensis*. These genotypes occurred differentially in the stroma, SFP (with ascocarps), and 2 types of ascospores of *C. sinensis* (*cf*. Table 2). The biomasses of the GC- and AT-biased genotypes undergo dynamic alterations in an asynchronous, disproportional manner during *C. sinensis* maturation, as demonstrated using a Southern blotting approach without DNA amplification [18, 27].

The ITS sequences of Genotype #1 were the most easily amplifiable and detectable using universal primers, but this finding may not necessarily imply dominance of the Genotype #1 DNA template in the test samples [7]. Southern blotting analysis without DNA amplification demonstrated that AT-biased genotypes were dominant in the stroma of natural *C. sinensis* during maturation, and GC-biased Genotype #1 was never the dominant fungus in the stroma [27, 28]. The easy pairing and elongation of the Genotype #1 ITS sequences using universal primers were observed due to (1) the sequence identity levels and (2) the absence of secondary structure/conformation in their sequences that could interfere with PCR amplification and sequencing [7]. In early molecular studies on *C. sinensis* (1999–2006), mycologists reported the detection of a single sequence of *H. sinensis* and therefore incorrectly judged *H. sinensis* to be the sole anamorph of *O. sinensis*.

Chen et al. [34] first reported the molecular heterogeneity of *C. sinensis*-associated fungi of the genera *Hirsutella*, *Paecilomyces* and *Tolypocladium* using a PCR amplicon cloning-sequencing methodology. However, insufficient attention was given to the "all-or-none" qualitative research technique and findings, and instead, the disproportionate amplicon clones selected for examination of the ITS sequences of different fungi were overemphasized, which unfortunately led to the improper conclusion that *H. sinensis* was the sole anamorph of *O. sinensis*. As demonstrated herein, we applied multiple pairs of genotype- and species-specific primers and cloning-based amplicon sequencing approaches with selection of at least 30 white colonies per Petri dish in cloning experiments, which enabled us to profile the components of the heterogeneous metagenome in the immature and mature stromata, SFP (with ascocarps), and ascospores of natural *C. sinensis*. In particular, this approach allowed us to identify those genotypes and fungi with low abundance and those with secondary structure/conformation in their DNA sequences, which may affect primer binding and DNA chain elongation during PCR amplification and sequencing [7, 10].

GC-biased Genotypes #1 and #2 coexisted in the stroma of natural *C. sinensis*, and their amplicon abundance was low in the immature stroma (1.0–2.0 cm in stromal height) [27, 28].

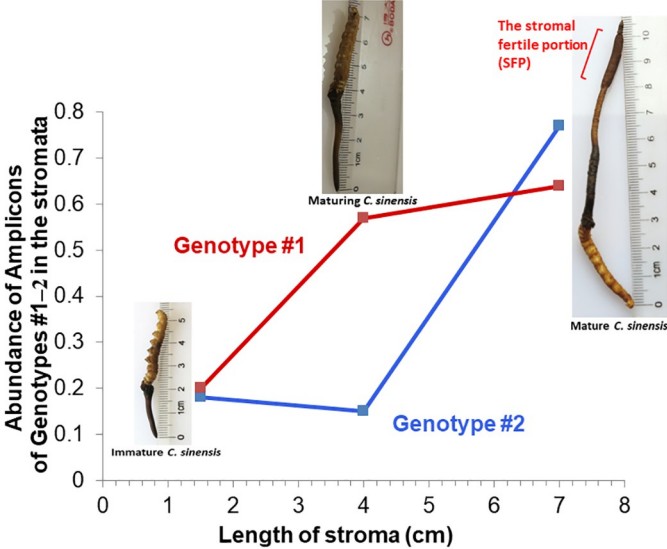

**Fig 8. Dynamic alterations of the abundance of the amplicons of Genotypes #1 and #2 of *O. sinensis* in the stromata of *C. sinensis* during maturation (modified from Fig 6 of [28]).** Immature *C. sinensis* had very short stroma of 1.5 cm. Maturing *C. sinensis* had stroma of 4.0 cm without an expanded fertile portion close to the stroma tip. Mature *C. sinensis* had long stroma of 7.0 cm and showed the formation of an expanded fertile portion close to the stroma tip, which was densely covered with ascocarps.

Genotype #1 showed a greatly elevated abundance in the maturing stroma (4.0–4.5 cm in height without an expanded fertile portion close to the stromal tip), and its abundance then plateaued in the mature stroma (6.5–7.0 cm in height, showing the formation of an expanded fertile portion close to the stromal tip, which is densely covered with ascocarps) (Fig 8; modified from Fig 6 of [28] in Chinese). In contrast to the maturation pattern of Genotype #1, the abundance of Genotype #2 remained at a low level in the maturing stroma and increased markedly in the mature stroma to a level that exceeded the abundance of Genotype #1. The cooccurrence of Genotypes #1 and #2 in the stroma with distinct maturation patterns indicates the genomic independence of the 2 GC-biased genotypes, which is proven by the absence of Genotype #2 sequences in the whole-genome sequences of 5 Genotype #1 *H. sinensis* strains [7–12, 49–53, 63, 64].

In this study, we did not detect GC-biased Genotypes #3 and #7–12 of *O. sinensis* in the compartments of natural *C. sinensis* specimens collected from the Hualong and Yushu areas of Qinghai Province. This may indicate that these GC-biased genotypes of *O. sinensis* may exist in the natural *C. sinensis* specimens in different production areas.

A Bayesian phylogenetic analysis (*cf.* Fig 7) demonstrated that the AT-biased genotypes were clustered into 2 branches, Clusters A and B. AT-biased genotypes from both branches were detected in the stroma and SFP of *C. sinensis*, but Genotypes #4 and #15 of Cluster-B were absent in the ascospores, consistent with those discovered by Li et al. [59]. Genotype #4 of AT-biased Cluster-B was the dominant AT-biased genotype in the immature stroma during the asexual growth stage and coexisted with less dominant Genotypes #5–6 of AT-biased Cluster-A [16, 17]. The abundance of Genotype #4 of Cluster-B markedly decreased in the mature *C. sinensis* stroma and remained at a low level in the SFP (with ascocarps) during the sexual growth stage. In contrast, the abundance of Genotype #5 of AT-biased Cluster-A increased reciprocally and significantly in the mature *C. sinensis* stroma and SFP and predominated in the *O. sinensis* ascospores [16, 17, 68]. In addition, Genotype #15 of Cluster-B predominated

in the SFP (with ascocarps) of *C. sinensis* prior to ascospore ejection and drastically declined after ejection [68]. The differential cooccurrence and alternative predominance of GC- and AT-biased genotypes of *O. sinensis* indicate their distinct physiological functions at different maturational stages of the *C. sinensis* lifecycle and their genomic independence, belonging to independent *O. sinensis* fungi.

Engh [30], Kinjo & Zang [32], Wei et al. [47] and Mao et al. [57] reported the detection of either Genotype #4 or #5 of *O. sinensis*, without codetection of GC-biased Genotype #1 *O. sinensis*, in some natural *C. sinensis* specimens collected from geographically distant areas and in cultivated *C. sinensis*. These findings likely indicate that the studied specimens might be in different maturational stages and that the amplicons of other cocolonized *O. sinensis* genotypes and fungal species might be overlooked during PCR amplification and sequencing due to lower-abundance amplicons caused by the experimental designs without using genotype- and species-specific primers or cloning-based amplicon sequencing strategies or due to secondary structures present in the sequences of *O. sinensis* genotypes, which may interfere with primer binding, DNA chain elongation, and sequencing [7].

In this study, we detected several AT-biased genotypes of Cluster-A and GC-biased Genotypes #1 *H. sinensis* and #13–14 in ascospores that cooccurred with *S. hepiali* and the AB067719-type fungus in the teleomorphic ascospores of natural *C. sinensis*. However, AT-biased Cluster-B genotypes were absent in ascospores but present in the SFP (with ascocarps). *Pseudogymnoascus roseus*, *Geomyces pannorum*, *Tolypocladium sinensis*, and *Penicillium chrysogenum* were not detected in the SFP (with ascocarps) and ascospores of *C. sinensis* despite their differential predominance in the stroma and caterpillar body of natural *C. sinensis* found in 2 previous mycobiota studies [38, 39].

The sequences of Genotypes #2–17 of *O. sinensis*, however, were not present in the genome assemblies (ANOV00000000, JAAVMX000000000, LKHE00000000, LWBQ00000000 and NGJJ00000000) of 5 Genotype #1 *H. sinensis* strains, Co18, IOZ07, 1229, ZJB12195 and CC1406-203, and instead belonged to the genomes of independent *O. sinensis* fungi [7–12, 27, 28, 49–54, 63, 64, 68]. These genomic independence findings support the hypotheses of independent *O. sinensis* fungi and an integrated microecosystem for natural *C. sinensis* [14, 24]. During the complex lifecycle of natural *C. sinensis*, its metagenomic members appear to function mutually and symbiotically at different development-maturation stages in a dynamically alternating manner from the immature to the maturing and then the mature stages and from initial asexual growth to sexual growth and reproduction.

## Complex anamorph-teleomorph connection of *O. sinensis*

The anamorph-teleomorph connection of *O. sinensis* has been the subject of decades-long academic debate [7–12, 29, 60]. Wei et al. [41] hypothesized that *H. sinensis* is the sole anamorph of *O. sinensis* based on the 3 aforementioned sets of evidence; this hypothesis satisfied the requirements of the first and second criteria of Koch's postulates, but unfortunately not the third and fourth criteria, making the hypothesis uncertain [7–12, 29, 54, 63, 64]. Although this hypothesis has been widely appreciated, Wei et al. [41] demonstrated significant genetic diversity of *O. sinensis* teleomorphs (*cf.* the far phylogenetic distance with a low bootstrap value between those of natural *C. sinensis* specimens G3 and S4, shown in Fig 6 of [41]).

Bushley et al. [75] discovered the multicellular heterokaryotic structure of the hyphae and ascospores of *C. sinensis*, which contain multiple mononucleated, binucleated and trinucleated cells. Li et al. [59], the research partners in the study [75], reported the identification of both GC-biased Genotype #1 and AT-biased Genotype #5 in 8 of 15 clones derived from the 25-day culture of a *C. sinensis* mono-ascospore. Based on insufficient and contradictory evidence,

these researchers overinterpreted that all AT-biased genotypes were "ITS pseudogene" components of the genome of Genotype #1 *H. sinensis*, in agreement with the sole anamorph hypothesis for *H. sinensis* [41]. However, AT-biased genotype sequences are not present in the 5 whole-genome assemblies (ANOV0000000, JAAVMX0000000000, LKHE00000000, LWBQ00 000000 and NGJJ00000000) of Genotype #1 *H. sinensis* strains Co18, IOZ07, 1229, ZJB12195 and CC1406-203 [7–10, 49–53, 64]. The genomic independence evidence disproved the "ITS pseudogene" hypothesis for the AT-biased genotypes of *O. sinensis* proposed by Li et al. [59].

In contrast, Barseghyan et al. [73] proposed a dual-anamorph hypothesis for *O. sinensis* involving *Tolypocladium sinensis* and *H. sinensis* using macro- and micromycological approaches, but it leaves an unresolved question regarding the genotype of *O. sinensis* with *H. sinensis*-like morphological and growth characteristics. Engh [30] hypothesized that *C. sinensis* and *T. sinensis* form a fungal complex in the natural insect-fungal complex, and the *C. sinensis* sequence AJ786590 was disclosed and uploaded to the GenBank database by Stensrud et al. [65]. Stensrud et al. [66] clustered AJ786590 and other sequences into Group-B of *O. sinensis* using a Bayesian phylogenetic approach and concluded that Group-B (AT-biased Genotype #4) sequences are phylogenetically distinct from GC-biased Group-A of *O. sinensis* (Genotype #1 *H. sinensis*).

Kinjo & Zang [32], Engh [30], Wei et al. [47] and Mao et al. [57] detected AT-biased Genotype #4 or #5 of *O. sinensis* from natural *C. sinensis* specimens collected from different geographic regions or from cultivated *C. sinensis*. Zhang et al. [55] and Cheng et al. [56] reported the detection of GC-biased Genotype #3 of *O. sinensis* in natural *C. sinensis* collected from Nyingchi in Tibet. Chen et al. [58] reported the detection of GC-biased Genotype #7 of *O. sinensis* from the stroma of *C. sinensis* specimens collected from Qinghai Province of China. These mutant genotypes of *O. sinensis* share the same *H. sinensis*-like morphological and growth characteristics without the cooccurrence of Genotype #1 *H. sinensis*, although they may share a common evolutionary ancestor [30, 32, 55–58, 66]. In the current study, we detected AT-biased Cluster-A genotypes (*cf.* Fig 7) and GC-biased Genotypes #1 and #13–14 in the teleomorphic ascospores of *C. sinensis*, along with *S. hepiali* and the AB067719-type fungus, and additional AT-biased Cluster-B genotypes in the stroma and SFP (with ascocarps) of *C. sinensis*.

Li et al. [48] recently reported the differential occurrence and transcription of the mating-type genes of the *MAT1-1* and *MAT1-2* idiomorphs in 237 *H. sinensis* strains, indicating genetic and transcriptional inability to perform self-fertilization under homothallic or pseudo-homothallic reproduction, as proposed by Hu et al. [49] and Bushley et al. [75]. Although Bushley et al. [75] reported the transcription of both the MAT1-1-1 and MAT1-2-1 genes of the *MAT1-1* and *MAT1-2* idiomorphs, the MAT1-2-1 transcript of *H. sinensis* strain 1229 harbored unspliced intron I, which contains 3 stop codons. This type of transcript presumably produces a truncated and dysfunctional MAT1-2-1 protein missing the majority portion of the protein encoded by exons II and III [48, 76]. These findings constitute a coupled transcriptional-translational mechanism of *H. sinensis* reproductive control, in addition to the controls at the genetic and transcriptional levels. The differential occurrence and transcription of the mating genes of the *MAT1-1* and *MAT1-2* idiomorphs indicate that *H. sinensis* requires a sexual partner to perform physiologically heterothallic reproduction in the lifecycle of natural *C. sinensis* regardless of whether *H. sinensis* is monoecious or dioecious [48, 76]. For example, *H. sinensis* strains 1229 and L0106 appear to reciprocally produce functional MAT1-1-1 and MAT1-2-1 transcripts and presumably complementary mating proteins [75, 77], possibly constituting a pair of sexual partners for physiological heterothallism [48, 76]. In addition, the transcripts of mating-type genes in *S. hepiali* strain Feng [78] were coincidentally found to be complementary to those of *H. sinensis* strain L0106 [77]. The coincident transcriptomic

findings reveal a possible transcriptomic mechanism for fungal hybridization, which triggers further investigation because *H. sinensis* is closely associated with a small quantity of *S. hepiali* in the compartments of natural *C. sinensis*, often resulting in difficulties in fungal isolation and purification even by top-notch mycology taxonomists [79].

Wei et al. [47] reported an industrial artificial cultivation project in which 3 anamorphic *H. sinensis* strains, 130508-KD-2B, 20110514 and H01-20140924-03, were reportedly used as inoculants. The successful cultivation of *C. sinensis* is important for supplementing the increasingly scarce natural resources of *C. sinensis* and for meeting the third criterion of Koch's postulates, adding academic value to supplement the 3 aforementioned sets of evidence [41]. However, these authors reported the identification of the sole teleomorph of *O. sinensis* in the fruiting body of cultivated *C. sinensis*, and it belonged to AT-biased Genotype #4. Thus, the apparent fungal mismatch between the inoculants used in artificial cultivation and the final cultivated product resulted in failure to meet the requirement of the fourth criterion of Koch's postulates. In addition to the detection of teleomorphic AT-biased Genotype #4 in cultivated *C. sinensis*, Wei et al. [47] also reported the detection of teleomorphic GC-biased Genotype #1 in natural *C. sinensis*. Because the sequence of AT-biased Genotype #4 is absent in the 5 whole-genome assemblies (ANOV00000000, JAAVMX000000000, LKHE00000000, LWBQ00000000 and NGJJ00000000) of the GC-biased Genotype #1 *H. sinensis* strains Co18, IOZ07, 1229, ZJB12195 and CC1406-203 [49–53], the scientific evidence reported by Wei et al. [47] disproves the sole anamorph and sole teleomorph hypotheses for *O. sinensis*, which were proposed 10 years ago by the same group of key authors [41]. The species mismatch reported by Wei et al. [47] may imply (1) that the researchers could have overlooked the Genotype #4 sequence in one or all of the anamorphic inoculant strains, which would confirm previous findings that GC-biased Genotype #1 *H. sinensis* (Group-A) and AT-biased genotypes (Groups B and C) formed a species complex [66] and that the actual causative agent is a fungal (species) complex containing several *O. sinensis* genotypes and *S. hepiali* [79]; or (2) that secondary or sequential infections by the true causal fungus/fungi (Genotype #4 and possibly other genotypes of *O. sinensis*) occurred in the course of artificial cultivation. A third possibility is that a series of preprogrammed, nonrandom mutagenic conversions of GC-biased Genotype #1 to AT-biased Genotype #4 occurred during artificial cultivation without exception in all cultivated *C. sinensis* products (and *vice versa* after the ejection of the ascospores); however, this possibility seems the least likely.

Zhang et al. [80] summarized nearly 40 years of artificial cultivation experience with failed induction of fruiting body and ascospore production in research-oriented academic settings, either in fungal cultures or after infecting insects with fungal inoculants. Distinct from the success of artificial *C. sinensis* cultivation in product-oriented industrial settings, as reported by Wei et al. [47], Hu et al. [49] inoculated 40 larvae of *Hepialus* sp. with 2 pure *H. sinensis* strains (Co18 and QH195-2) *via* the injection of a mycelial mixture through the second larval proleg, which caused death and mummification of the larvae but failed to induce the production of a fruiting body. Li et al. [79] inoculated 400 larvae of *Hepialus armoricanus* with 4 groups of inoculants (n = 100 larvae per inoculant): (1) conidia of *H. sinensis*, (2) mycelia of *H. sinensis*, (3) purified ascospores of *C. sinensis*, and (4) a mixture of 2 wild-type fungal strains, CH1 and CH2, isolated from the intestines of healthy living larvae of *Hepialus lagii* Yan [81]. Both strains CH1 and CH2 showed *H. sinensis*-like morphological and growth characteristics and contained GC-biased Genotype #1 *H. sinensis* and *S. hepiali* as well as highly abundant AT-biased Genotype #5 of *O. sinensis* and the markedly less abundant Genotypes #4 and #6 [79]. The application of a mycelial mixture of strains CH1 and CH2 as the inoculant resulted in a favorable infection-mortality-mummification rate of 55.2±4.4%, indicating 15–39-fold greater potency than the infection-mortality-mummification rates (1.4–3.5%; P<0.001) achieved after

inoculation with the conidia or mycelia of *H. sinensis* or ascospores of *C. sinensis*. These findings reported by Li et al. [48, 79], Wei et al. [47], Hu et al. [49], and Zhang et al. [80] suggest that GC-biased Genotype #1 *H. sinensis* may not be the sole true causative fungus in natural *C. sinensis* and that inoculation synergy of the symbiotic fungal species may be needed to initiate the development of the stromal primordia and fruiting bodies, and sexual partners, regardless of whether they are the same or interspecific species, may be needed to induce the transition from asexual to sexual growth and reproduction and the maturation of the teleomorphic ascocarps and ascospores during the lifecycle of natural and cultivated *C. sinensis*.

The microcycle conidiation of *C. sinensis* ascospores using culture protocols favoring the growth of *H. sinensis* has also been reported [42–44]. Bushley et al. [75] demonstrated that *C. sinensis* ascospores are multicellular and heterokaryotic, consisting of multiple monokaryotic, bikaryotic and trikaryotic cells. Our study herein demonstrated that the fully ejected *C. sinensis* ascospores contain GC-biased Genotypes #1 and #14, AT-biased Genotypes #5–6 and #16 within AT-biased Cluster-A, *S. hepiali* and the AB067719-type fungus (*cf.* Table 2 and Fig 7). These findings suggest a multicellular heterokaryotic structure and genetic heterogeneity of ascospores [7–10]. However, the culture-dependent approach detected only GC-biased Genotype #1 and AT-biased Genotype #5 after 25-day liquid inoculation of mono-ascospores [59], suggesting the possibility of overlooking several AT-biased genotypes and GC-biased Genotypes #14, probably due to nonculturability of those genotypes or inappropriately designed and executed molecular techniques. The nonculturable nature of most genotypes of *O. sinensis* fungi examined to date [60–62] calls into question whether all genotypes of *O. sinensis* in all cells of multicellular heterokaryotic ascospores are capable of undergoing conidiation through *in vitro* microcycle conidiation under the commonly used experimental conditions that favor the growth of Genotype #1 *H. sinensis*. The lack of molecular information on conidia in these studies [42–44] makes the fungal/genotypic identity of the conidia uncertain due to the *H. sinensis*-like morphological and growth characteristics shared by multiple genotypes of *O. sinensis* [30, 32, 47, 55–59, 75, 79] and the *Hirsutella*-like morphology shared by numerous fungal species in the families Ophiocordycipitaceae and Clavicipitaceae and the genera *Polycephalomyces* and *Harposporium* [82].

The study described herein further revealed fungal and genotypic heterogeneity in the multicellular heterokaryotic ascospores of *C. sinensis*. AT-biased Cluster-A genotypes, GC-biased Genotype #1 *H. sinensis* and Genotypes #13 or #14 (showing large DNA segment reciprocal substitutions and genetic material recombination) in either type of ascospores, *S. hepiali* and the AB067719-type Group-E fungus naturally cooccur in teleomorphic *C. sinensis* ascospores.

## Differential occurrence of Genotypes #4 and #15 of AT-biased Cluster-B

Studies of natural and cultivated *C. sinensis* have detected Genotype #4 of *O. sinensis*, but not coexisting GC- and AT-biased genotypes of *O. sinensis*, including AJ786590 discovered by Engh [30] and uploaded to GenBank by Stensrud et al. [65], AB067741–AB067749 identified by Kinjo & Zang [32], KC305891–KC305892 discovered by Mao et al. [57], and 2 sequences ("Fruiting body 3" and "Mycelia 3", sequences unavailable in GenBank) with high homology to AB067749 and KC305892 found by Wei et al. [47]. These studies did not report the maturation stages of natural and cultivated *C. sinensis* specimens that were used as the study materials. However, other studies using genotype-specific primers, amplicon cloning and biochip-based SNP mass spectrometry genotyping techniques reported the detection of Genotype #4 of *O. sinensis* in natural *C. sinensis* coexisting with Genotype #1 *H. sinensis* with or without Genotypes #5–6 [16–18, 24, 27, 28, 59, 68].

Genotype #4 of AT-biased Cluster-B was found to predominate in the stroma of immature *C. sinensis* and to decline during *C. sinensis* maturation (*cf.* Fig 7) [16, 17, 27, 28, 63, 64, 68].

Genotype #4 and the newly discovered Genotype #15 of AT-biased Cluster-B are present in the SFP (with ascocarps) but not in ascospores of natural *C. sinensis* (*cf*. Table 2) [68].

## Fungal factors involved in the control of the development, maturation and ejection of *C. sinensis* ascospores

In this study, we report the discovery of 2 types of ascospores of *C. sinensis*, namely, fully and semiejected ascospores. In contrast to the well-described fully ejected ascospores, semiejected ascospores show tight adhesion to the outer surface of asci, hanging out of the perithecial opening (*cf*. Figs 1C, 4B and 4E). The distinct ejection behaviors of ascospores are associated with divergent fungal compositions. In addition to the cooccurrence of GC-biased Genotype #1, AT-biased Cluster-A genotypes and the AB067719-type fungus in the 2 types of ascospores, the semiejected ascospores contain Genotype #13 of *O. sinensis* and a greater abundance of *S. hepiali*, whereas the fully ejected ascospores contain Genotype #14 and a markedly lower abundance of *S. hepiali* (*cf*. Fig 5 and Table 2). Genotypes #13–14 show alternating reciprocal substitutions of large DNA segments between the genomes of 2 parental fungi, namely, Genotype #1 *H. sinensis* (Group-A by [66]) and the AB067719-type Group-E fungus (*cf*. Table 3 and Fig 6), indicating that the biological processes of plasmogamy, karyogamy, chromosomal intertwining interaction, and genetic material recombination occur differentially between the 2 parental fungi regardless of the processes of hereditary variation caused by fungal hybridization or parasexual reproduction [9, 48]. The divergent fungal components with altered abundances may participate in the control of the development, maturation and ejection of *C. sinensis* ascospores.

Genotypes #6 and #16 of AT-biased Cluster-A were detected in fully ejected ascospores but not in semiejected ascospores using genotype-specific primers and cloning-based amplicon sequencing (*cf*. Table 2; Fig 7). The differential occurrence of these 2 AT-biased genotypes may need to be further verified due to the possible existence of secondary structures/conformation in the DNA sequences that may affect primer binding and DNA chain elongation during PCR amplification and sequencing [7–10]. In fact, Genotypes #6 and #16 were detected in both types of ascospores using the biochip-based MassARRAY SNP mass spectrometry technique [68].

## Conclusions

This study identified 2 types of multicellular heterokaryotic ascospores collected from the same natural *C. sinensis* specimens. Multiple genotypes of *O. sinensis* coexist differentially in the 2 types of *C. sinensis* ascospores and are accompanied by *S. hepiali* and the AB067719-type fungus. The divergent fungal components of the 2 types of ascospores may participate in the control of the development, maturation and ejection of the ascospores. The AT-biased Cluster-A genotypes are present in the stroma, SFP (with ascocarps), and ascospores; however, the AT-biased Cluster-B genotypes are present in the stroma and SFP but not in ascospores. Multiple fungal components occur differentially in the compartments of natural *C. sinensis* and undergo dynamic alterations in an asynchronous, disproportional manner during *C. sinensis* maturation. These findings describe part of the complex lifecycle of this precious TCM therapeutic agent whose metagenomic components undergo dynamic alterations at different development and maturation stages in a symbiotic manner.

## Supporting information

**S1 File.**
(ZIP)

**S1 Fig.**
(TIF)

## Acknowledgments

The authors are grateful to Prof. Mu Zang, Prof. Ru-Qin Dai, Prof. Ying-Lan Guo, Prof. Ping Zhu, Prof. Zong-Qi Liang, Prof. Zhao-Lan Li, Prof. Yu-Guo Zheng, Dr. Jia-Gang Zhao and Dr. Yan-Jiao Zhou for consultations and Prof. Xin Liu, Prof. Hai-Feng Xu, Ms. Xiao-Li Ma, Ms. Ming Yang, Mr. Zong-Hao Zhang, Mr. Wei Chen, Mr. Tao-Ye Zheng, Mr. Jin-Jin Li, and Mr. Yu-Chun Zhou for their assistance.

## Author Contributions

**Conceptualization:** Yu-Ling Li, Wei-Dong Xie, Jian-Yong Wu, Jia-Shi Zhu.

**Data curation:** Xiu-Zhang Li, Yi-Sang Yao, Ling Gao, Ning-Zhi Tan.

**Funding acquisition:** Yu-Ling Li, Jia-Shi Zhu.

**Investigation:** Xiu-Zhang Li, Yi-Sang Yao, Zi-Mei Wu, Ling Gao, Ning-Zhi Tan, Zhou-Qing Luo, Jian-Yong Wu.

**Methodology:** Yu-Ling Li, Ling Gao, Zhou-Qing Luo.

**Project administration:** Wei-Dong Xie.

**Resources:** Yu-Ling Li, Yi-Sang Yao, Zi-Mei Wu, Ning-Zhi Tan.

**Supervision:** Jia-Shi Zhu.

**Validation:** Xiu-Zhang Li, Yi-Sang Yao, Ling Gao.

**Visualization:** Xiu-Zhang Li.

**Writing – original draft:** Jia-Shi Zhu.

**Writing – review & editing:** Jia-Shi Zhu.

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
