## [Decision Letter · Decision Letter 0]

2 Aug 2022

PONE-D-22-17339Differential coexistence of multiple genotypes of Ophiocordyceps sinensis in the stromata, ascocarps and ascospores of natural Cordyceps sinensisPLOS ONE

Dear Dr. Zhu,

Thank you for submitting your manuscript to PLOS ONE. After careful consideration, we feel that it has merit but does not fully meet PLOS ONE’s publication criteria as it currently stands. Therefore, we invite you to submit a revised version of the manuscript that addresses the points raised during the review process. The paper has been revised by two experts who bith sufggested some modifications to be included inthe revised version of the ms in order to improve it and render the work acceptable for publication.

We look forward to receiving your revised manuscript.

Kind regards,

Sabrina Sarrocco

Academic Editor

PLOS ONE

Journal Requirements:

"This research was supported by a grant from the Science and Technology Department of Qinghai Province, China, grant number 2021-SF-A4 “Study on key technologies of conservation of natural resource and industrial upgrading of Cordyceps sinensis”, the major science and technology projects in Qinghai Province."

 "This research was supported by a grant from the Science and Technology Department of Qinghai Province, China, grant number 2021-SF-A4 “Study on key technologies of conservation of natural resource and industrial upgrading of Cordyceps sinensis”, the major science and technology projects in Qinghai Province."

Reviewers' comments:

Reviewer's Responses to Questions

**Comments to the Author**

1. Is the manuscript technically sound, and do the data support the conclusions?

Reviewer #1: Partly

Reviewer #2: Partly

2. Has the statistical analysis been performed appropriately and rigorously? 

Reviewer #1: Yes

Reviewer #2: N/A

3. Have the authors made all data underlying the findings in their manuscript fully available?

Reviewer #1: Yes

Reviewer #2: Yes

4. Is the manuscript presented in an intelligible fashion and written in standard English?

Reviewer #1: Yes

Reviewer #2: No

5. Review Comments to the Author

Reviewer #1: Cordyceps sinensis is a famous-medicinal fungi, and has various function on many dieseases. This aticle reports mutiple genotype coexistence at differentially developmental stages, and has significant meaning. However, some questions and the latin names need be revised.

Reviewer #2: In this ms, immature and mature C. sinensis were harvested, and mature C. sinensis were cultivated in laboratory to collect ascospores. Fully and semi-ejected

ascospores were collected from the same sample. The molecular analyses species-/genotype-specific primers were used to analyze multiple genotypes of Ophiocordyceps sinensis in the stromata, ascocarps and ascospores of natural Cordyceps sinensis. The results indicated the metagenomic components undergo dynamic change at different development stages in a symbiotic manner, which would help to understand the development, the maturation of ascospores and artificial cultivation.

However, a number of points need to be clarified and further justified, given below,

The abstract is too simple to set out the main opinion of this paper.

Semi-ejected ascospore was considered to be one type that differ from the ejected ascospore. But be lack of the evidence in morphological and growth characteristics.

In instruction part, the author didn’t provide detailed and clear background information, and the description and the definitions are confused, e.g. the stromal fertile portion (SFP), C. sinensis , O. sinensis, natural C. sinensis , wild C. sinensis. O. sinensis fungus, the statement is wrong.

Most sentences contain grammar mistakes. e.g. line 62-70, ‘ a dead Hepialidae moth larva’, should be the remains of Hepialidae moth larva,

‘intact intestine and head tissues, an intact, thick larval body cultivated insect-fungal complex’... The expressions are redundant and very difficult to read. There are many similar problems in this part.

In method part, the statement is too simple. Be deficient in descriptions of agents, materials, protocols, etc reagents, materials, protocol. And references are not fully cited.

e. g. before the extraction, how the samples were stored? How to extract?

What gene was genotype primers designed with？Whether ITS1-5.8S-ITS2 segment is suitable for all the three types primers ? Why was touch-down PCR program used? ...

Line 120-121,‘ the collected samples were cultivated in our laboratory (altitude 2,200 m) in Xining City, Qinghai Province of China, with the windows left open and under conditions with sufficient watering, sunshine and ventilation.’ The culturing conditions present unclear, including light, water,ventilation,soil..., The collection method of fully/semi- ejected ascospores present unclear.

The part of result and discussion is redundant and too long, and lack a sense of reading and hierarchy. Suggested to thoroughly revise.

References

The references weren’t be completely correct and need to be formatted according to the requirement of Plos one.

Grammar mistakes,

It is very noted that this manuscript needs careful editing by expertise in English editing in pay attention to the spelling, grammar, sentence structure. Otherwise, the content can’t be clearly present. Most sentences contains grammar mistakes in this ms.

6. PLOS authors have the option to publish the peer review history of their article (what does this mean?). If published, this will include your full peer review and any attached files.

Reviewer #1: No

Reviewer #2: No

---

## [Author Response · Author response to Decision Letter 0]

14 Sep 2022

Please see the submitted file "2022-9-13 Responses to the reviewers.docx" for all the responses to the reviewers' comments.

---

## [Decision Letter · Decision Letter 1]

10 Feb 2023

PONE-D-22-17339R1Differential coexistence of multiple genotypes of Ophiocordyceps sinensis in the stromata, ascocarps and ascospores of natural Cordyceps sinensisPLOS ONE

Dear Dr. Zhu,

Thank you for submitting your manuscript to PLOS ONE. The reviewers have appreciated the improvements made and only some figures have to be optimised and the references have to be edited. Therefore, we invite you to submit a revised version of the manuscript that addresses the points raised during the review process by reviewer #2.

We look forward to receiving your revised manuscript.

Kind regards,

Olaf Kniemeyer

Academic Editor

PLOS ONE

Journal Requirements:

Reviewers' comments:

Reviewer's Responses to Questions

**Comments to the Author**

1. If the authors have adequately addressed your comments raised in a previous round of review and you feel that this manuscript is now acceptable for publication, you may indicate that here to bypass the “Comments to the Author” section, enter your conflict of interest statement in the “Confidential to Editor” section, and submit your "Accept" recommendation.

Reviewer #1: All comments have been addressed

Reviewer #2: (No Response)

2. Is the manuscript technically sound, and do the data support the conclusions?

Reviewer #1: Yes

Reviewer #2: Yes

3. Has the statistical analysis been performed appropriately and rigorously? 

Reviewer #1: Yes

Reviewer #2: Yes

4. Have the authors made all data underlying the findings in their manuscript fully available?

Reviewer #1: Yes

Reviewer #2: Yes

5. Is the manuscript presented in an intelligible fashion and written in standard English?

Reviewer #1: Yes

Reviewer #2: Yes

6. Review Comments to the Author

Reviewer #1: (No Response)

Reviewer #2: In this manuscript, the authors concluded that multiple genotypes of O. sinensis coexisted differentially in the stromata, SFPs and 2 types of C. sinensis ascospores, along with S. hepiali and the AB067719-type fungus by molecular analyses. The results indicated the fungal components in different combinations and their dynamic alterations in the compartments of C. sinensis during maturation played symbiotic roles in the lifecycle of natural C. sinensis. This study would help to understand the development, the maturation of ascospores and artificial cultivation. However, some questions need be revised.

1. The bar in Figure 4A is not clear.

2. Modify the Bayesian tree in Figure 7. Some words are stacked together, and arrow marks are easy to misunderstanding.

3. Please unify the format of references, e.g. 74-76, 83 inconsistent with others.

7. PLOS authors have the option to publish the peer review history of their article (what does this mean?). If published, this will include your full peer review and any attached files.

Reviewer #1: No

Reviewer #2: No

---

## [Author Response · Author response to Decision Letter 1]

16 Feb 2023

Responses to the comments from reviewer #2

Reviewer #2: In this manuscript, the authors concluded that multiple genotypes of O. sinensis coexisted differentially in the stromata, SFPs and 2 types of C. sinensis ascospores, along with S. hepiali and the AB067719-type fungus by molecular analyses. The results indicated the fungal components in different combinations and their dynamic alterations in the compartments of C. sinensis during maturation played symbiotic roles in the lifecycle of natural C. sinensis. This study would help to understand the development, the maturation of ascospores and artificial cultivation. However, some questions need be revised.

1. The bar in Figure 4A is not clear.

2. Modify the Bayesian tree in Figure 7. Some words are stacked together, and arrow marks are easy to misunderstanding.

3. Please unify the format of references, e.g. 74-76, 83 inconsistent with others.

Authors’ response:

1. The bar in Figure 4A is not clear.

We have modified Figure 4A with a thicker bar and an enlarged font.

2. Modify the Bayesian tree in Figure 7. Some words are stacked together, and arrow marks are easy to misunderstanding.

We have rearranged the arrows and marks in red and blue in Figure 7 to not stack the tree lines and bootstrapping values.

However, as we stated in the Methods section “Phylogenetic analysis of ITS sequences”, the Bayesian majority-rule consensus tree was inferred using MrBayes v3.2.7a software (the Markov chain Monte Carlo [MCMC] algorithm by Nanjing Genepioneer Biotechnologies Co. as a commercial technical service, because we have failed using our personal computers that were not able to perform such computational works. The tree lines and bootstrapping values in black in the tree were calculated using the company’s computer and software with a sampling frequency of 103 iterations after discarding the first 25% of samples from a total of 1.1x108 iterations. We are unfortunately not able to change the location of bootstrapping values in the Bayesian tree.

3. Please unify the format of references, e.g. 74-76, 83 inconsistent with others.

We have carefully checked the entire reference list and made corrections.

---

## [Editor Report · Decision Letter 2]

20 Feb 2023

Differential coexistence of multiple genotypes of Ophiocordyceps sinensis in the stromata, ascocarps and ascospores of natural Cordyceps sinensis

PONE-D-22-17339R2

Dear Dr. Zhu,

We’re pleased to inform you that your manuscript has been judged scientifically suitable for publication and will be formally accepted for publication once it meets all outstanding technical requirements.

Kind regards,

Olaf Kniemeyer

Academic Editor

PLOS ONE

Additional Editor Comments (optional):

All critical points have been addressed.
---

## [Editor Report · Acceptance letter]

28 Feb 2023

PONE-D-22-17339R2 

Differential coexistence of multiple genotypes of *Ophiocordyceps sinensis* in the stromata, ascocarps and ascospores of natural *Cordyceps sinensis*

Dear Dr. Zhu:

I'm pleased to inform you that your manuscript has been deemed suitable for publication in PLOS ONE. Congratulations! Your manuscript is now with our production department. 

Kind regards, 

on behalf of

Dr. Olaf Kniemeyer 

Academic Editor

PLOS ONE